# Impulsive Personality Traits Predicted Weight Loss in Individuals with Type 2 Diabetes after 3 Years of Lifestyle Interventions

**DOI:** 10.3390/jcm11123476

**Published:** 2022-06-16

**Authors:** Giulia Testa, Lucía Camacho-Barcia, Carlos Gómez-Martínez, Bernat Mora-Maltas, Rafael de la Torre, Xavier Pintó, Dolores Corella, Roser Granero, Aida Cuenca-Royo, Susana Jiménez-Murcia, Nancy Babio, Rebeca Fernández-Carrión, Virginia Esteve-Luque, Laura Forcano, Jiaqi Ni, Mireia Malcampo, Sara De las Heras-Delgado, Montse Fitó, Jordi Salas-Salvadó, Fernando Fernández-Aranda

**Affiliations:** 1Psychoneurobiology of Eating and Addictive Behaviors Group, Institut d’Investigació Biomèdica de Bellvitge (IDIBELL), 08907 Barcelona, Spain; gtesta@idibell.cat (G.T.); lcamacho@idibell.cat (L.C.-B.); bernatmoramaltas@gmail.com (B.M.-M.); roser.granero@uab.cat (R.G.); sjimenez@bellvitgehospital.cat (S.J.-M.); 2Department of Psychiatry, University Hospital of Bellvitge, 08907 Barcelona, Spain; 3CIBER Physiology of Obesity and Nutrition (CIBEROBN), Carlos III Health Institute, 28029 Madrid, Spain; carlos.gomez@urv.cat (C.G.-M.); rtorre@imim.es (R.d.l.T.); dolores.corella@uv.es (D.C.); acuenca@imim.es (A.C.-R.); nancy.babio@urv.cat (N.B.); rebeca.fernandez@uv.es (R.F.-C.); lforcano@imim.es (L.F.); mfito@imim.es (M.F.); 4Universitat Rovira i Virgili, Departament de Bioquímica i Biotecnologia, Unitat de Nutrició Humana, 43201 Reus, Spain; jiaqi.ni@alumni.urv.cat (J.N.); sara.delasheras@urv.cat (S.D.l.H.-D.); 5Institut d’Investigació Sanitària Pere Virgili (IISPV), Unitat de Nutrició Humana, 43201 Reus, Spain; 6Neurosciences Program, Institut Hospital del Mar d’Investigacions Mèdiques (IMIM), 08003 Barcelona, Spain; 7Medicine and Life Sciences Department, Universitat Pompeu Fabra (UPF), 08002 Barcelona, Spain; 8Lipids and Vascular Risk Unit, Internal Medicine, University Hospital of Bellvitge-IDIBELL, Hospitalet de Llobregat, 08907 Barcelona, Spain; xpinto@bellvitgehospital.cat (X.P.); virginia.esteveluque@gmail.com (V.E.-L.); 9Department of Clinical Sciences, School of Medicine and Health Sciences, University of Barcelona, Hospitalet de Llobregat, 08907 Barcelona, Spain; 10Department of Preventive Medicine, University of Valencia, 46010 Valencia, Spain; 11Department of Psychobiology and Methodology, Autonomous University of Barcelona, 08193 Barcelona, Spain; 12Unit of Cardiovascular Risk and Nutrition, Hospital del Mar Institute for Medical Research (IMIM), 08003 Barcelona, Spain; mireiamalcampo@gmail.com

**Keywords:** impulsivity, type 2 diabetes, weight loss, glycated hemoglobin

## Abstract

Impulsivity has been associated with type 2 diabetes (T2D) and may negatively impact its management. This study aimed to investigate impulsive personality traits in an older adult population with T2D and their predicting role in long-term weight control and glycemic management, through glycated hemoglobin (HbA_1c_), following 3 years of intervention with a Mediterranean diet. The Impulsive Behavior Scale (UPPS-P) was administered as a measure of impulsive traits at baseline. Results showed higher total baseline scores of UPPS-P, and higher positive urgency in individuals with T2D, compared with those without T2D. The regression analysis in patients with T2D showed that sensation seeking and lack of perseverance predicted weight loss at follow-up. By contrast, impulsive traits did not predict follow-up levels of HbA_1c_. In conclusion, the present findings suggest that higher impulsive traits in individuals with T2D seem to affect long-term weight control, but not glycemic control.

## 1. Introduction

Type 2 Diabetes (T2D) is a metabolic disorder that represents a major concern in the global healthcare system, as its prevalence has rapidly increased worldwide. This increment is especially observable in older adults, aged 65 or older, who constitute about half of the total population with this condition [1]. Several factors associated with the ageing process increase the susceptibility to T2D, including the reduced β-cell sensitivity to glucose, diminution of skeletal muscle tissue and increment of central adiposity [2]. The presence of T2D increases the risk of several metabolic comorbidities, as well as the development of cognitive decline [3], factors that constitute a challenge in the management of T2D [4]. 

Impulsive behaviors, defined as the tendencies to act rashly without giving adequate forethought to the consequences of the behaviors, could contribute to diabetes mismanagement in terms of diet, physical activity, medication adherence and blood sugar control. So far, impulsivity has been conceived as a multidimensional construct, which includes personality traits and behavioral and cognitive factors [5]. Obesity has been strongly linked to impulsivity at several levels, and the underlying mechanisms have begun to be elucidated [6,7,8,9,10,11,12]. In terms of impulsivity-related cognitive functions, studies in individuals with T2D have shown poor decision-making and reduced inhibitory control [13,14], possibly due to the impact of insulin resistance and impaired glycemic control on the brain dopaminergic pathways [15,16].

Impulsivity has also been conceptualized in terms of more stable traits by multiple models of personality [5]. Previous studies that explored personality traits in individuals with T2D [17,18,19] mostly adopted the Big Five factors model of personality [20], which is a global measure of personality that might be suboptimal for exploring impulsive traits. A widely accepted instrument used to conceptualize the multivariable structure of impulsivity is the impulsive behavior scale (UPPS) [21], which includes five dimensions related to various impulsive behaviors (i.e., sensation seeking, negative urgency, positive urgency, lack of premeditation and lack of perseverance). To date, no studies have been conducted in patients with T2D using the UPPS model, which could help to better characterize impulsive traits in this population. Considering the evidence for impulsivity-related cognitive dysfunctions shown in the individuals with T2D [13,14], it could also be expected to find further impulsive personality traits in these individuals. 

Given that impulsivity leads to a lack of planning and insufficient regulation of behavior, impulsive traits may be expected to predispose to poor weight control and reduced self-management of T2D in the long term. In line with this view, a recent study in patients with T2D showed a negative correlation between impulsive traits and diabetes self-management, and a positive correlation with self-reported glycated hemoglobin (HbA_1c_) levels, an indicator of glycemic control [22]. In addition, impulsive personality has been shown to indirectly and negatively predict exercise and diet adherence via diabetes management self-efficacy [23]. Nevertheless, the cross-sectional designs of these studies did not provide a conclusion as to whether the impulsive personality could have a long-term impact on diabetes self-management.

Regarding the relationship between BMI and impulsive traits in T2D, a study in a large population of individuals with T2D showed that higher scores in the Barratt Impulsiveness Scale (BIS-11) were related to higher BMI [24]. Still, no studies have explored the relationship between impulsive traits and difficulties in long-term weight control in individuals with T2D. However, in a recent study performed in an older adult population with obesity, individuals with lower impulsive traits achieved higher BMI decreases at one-year follow-ups [25]. Longitudinal studies are needed in patients with T2D to elucidate the impact of impulsive personality traits on long-term weight management.

The present study first aimed to compare impulsive personality traits between older Mediterranean adult individuals with obesity in the presence or absence of T2D, and to explore the relationship between impulsivity and diabetic metabolic indicators among the population with T2D. 

A second aim was to identify in patients with T2D whether impulsive personality traits were predictors of weight loss and diabetes control after 3 years of lifestyle interventions. We hypothesized that patients with T2D would be characterized by more impulsive personality traits, according to the UPPS model, and impulsivity was expected to negatively affect weight loss and glycemic control after an intervention with a Mediterranean diet (MedDiet). 

## 2. Materials and Methods

### 2.1. Study Design and Participants

This longitudinal analysis was performed in the framework of the PREDIMED-Plus-Cognition sub-study (*n* = 487) (see Appendix A. Flowchart for the sampling, Appendix A), a subset sample of the clinical trial PREDIMED-Plus, a large, multicentric, randomized study (*n* = 6874) that aimed to assess the effect of an energy-restricted Mediterranean diet, physical activity promotion and behavioral intervention on the primary prevention of cardiovascular disease. The design, procedure and population was reported previously [26], and its full protocol is available in the RCT website (https://predimedplus.com, accessed on 1 April 2022). The trial was registered at the International Standard Randomized Controlled Trial in the year 2014, with the number ISRCTN89898870. Criteria for selecting the subjects were as follows: community-dwelling adults with Metabolic Syndrome (with any 3 of 5 risk factors [27]), aged between 55 and 75 years, with a body mass index (BMI) between 27 and 40 kg/m^2^. Participants with diabetes constituted approximately 25% of the final total PREDIMED-Plus sample. Individuals were recruited from 23 Spanish health centers that participated in the study, between October 2013 and December 2016, and randomly assigned to either the intervention group or the control group. The intervention is based on an energy restricted MedDiet, with behavioral support and physical activity. The individuals randomized into the control group were advised to follow an unrestricted MedDiet without any further indication.

### 2.2. Procedure

The Cognition sub-study had a specific objective, to perform an in-depth assessment of psychological and cognitive performance in this sub-sample. For this purpose, four of the recruiters’ sites: Bellvitge University Hospital (Barcelona); Universitat Rovira i Virgili (Reus); University of Valencia (Valencia); and Hospital del Mar Medical Research Institute (Barcelona); assessed cognitive function and trait impulsivity through a battery of questionnaires and tests at baseline (T0) and at three years of follow-up (T3). During these visits, investigators also collected additional information, such as clinical and anthropometric variables. 

### 2.3. Measures

#### 2.3.1. Anthropometric and Metabolic Measurements

At baseline, and in each yearly follow-up visit thereafter, trained personnel measured different anthropometric outcomes, including weight and height, that were then used to establish the body mass index [BMI = weight(kg)/height(m)^2^]. The BMI was classified using the World Health Organization (WHO)’s nutritional status ranges [28]. Fasting blood samples were taken at baseline, 1-year and 3-year of follow-up to determine the levels of fasting blood glucose, glycated hemoglobin (HbA_1c_), insulin and lipid profile. Insulin levels measured at baseline were used to estimate the Homeostasis Model Assessment of Insulin Resistance index (HOMA-IR= fasting glucose levels [mg/dL] × fasting insulin levels [µU/mL]/405.13) [29]. 

#### 2.3.2. Trait Impulsivity Assessment

This study used the UPPS-P framework [21], assessing five impulsive behavior pathways to measure impulsivity traits: (1) sensation seeking, described as the tendency to seek sensory pleasure and excitement; (2) lack of premeditation, described as the tendency to act without forethought and planning; (3) lack of perseverance, defined as the tendency to not finish tasks; (4) negative urgency, defined as the tendency to act rashly in negative emotional states; and (5) positive urgency, defined as the tendency to act rashly in positive emotional states. Each item of the UPPS-P is rated on a 4-point Likert scale ranging from 1—“strongly agrees” to 4—“strongly disagree”. Scores for each dimension and a total score of the UPPS-P were calculated, with higher scores indicating higher levels of impulsivity. Previous studies have shown that these traits share between 6% and 27% of their variance, with negative and positive urgency sharing the largest proportion of variance [30]. The measurement of separate aspects of impulsivity can clarify discrete relationships that might be masked or watered-down when such constructs are combined [31]. The Spanish adaptation of this instrument, which was used in the present study, showed adequate psychometric properties [32]. In our sample, internal consistency ranged from adequate (α = 0.75 for lack of perseverance) to excellent (α = 0.92 for the negative urgency subscale). 

### 2.4. Statistical Analysis 

Stata17.0 was used for the statistical analysis [33]. The comparison between the groups (presence/absence of T2D) at baseline (T0) was based on chi-square tests (χ^2^) for categorical variables and T-tests for independent samples for continuous variables. Partial correlation coefficients (adjusted by sex, age and education levels) assessed the association at baseline (T0) between the impulsive measures (UPPS-P) and the insulin-related metabolic conditions. Multiple regression was used to assess the predictive capacity of the impulsivity levels (UPPS-P scores) at baseline (T0) on the BMI, weight and HbA_1c_ measures at the 3-year follow-up (T3). These regressions were performed in two blocks: (a) in the first block, the covariates sex, age, education and intervention group were entered and fixed (ENTER method); (b) in the second block, the significant predictors of the criteria were automatically selected by the stepwise method (this selection was used due the large set of potential predictors and the exploratory nature of the analysis) [34]. The final models generated by the regression procedures were next considered adequate if they allowed for reliable clinical interpretation. The ANOVA and regression models were adjusted by sex, age, education and the intervention group. The level of significance was *p* < 0.05.

## 3. Results

### 3.1. Description of the Sample 

Table 1 contains the descriptive for the total sample at baseline, as well as the comparison of the participants with (T2D+) and without (T2D−) diabetes at baseline. No differences between the groups were found for the sociodemographic characteristics, but the participants in the T2D+ group were older and also had higher levels of HOMA-IR, glucose and HbA_1c_. Regarding the UPPS-P scores, positive urgency and total UPPF-P score were higher within T2D+ participants.

### 3.2. Association of Impulsivity with Markers of Glucose Metabolism

No relevant associations were found between the UPPS-P scores with the HOMA-IR, plasma-fasting glucose or HbA_1c_ levels at baseline. All of the partial correlations obtained non-significant results (*p* < 0.05), and coefficients were within the low-effect size range (in the study, the |R| were between 0.003 and 0.098) (see Appendix A).

### 3.3. Predictive Capacity of the Impulsivity at Baseline on BMI, Weight and HbA_1c_

Table 2 shows the results for the final regressions obtained in the multiple regressions, using the subsample of participants with T2D. Higher scores in the lack of perseverance subscale at baseline were associated with higher BMI levels at the end of the study, while a higher total score of UPPS-P predicted a higher weight at year 3. Higher scores of sensation seeking predicted lower decreases in the BMI and the weight. The impulsivity levels at baseline were not associated with the HbA_1c_ levels at 3 years of follow-up (mean = 6.74, SD = 1.00), nor with the decrease in the HbA_1c_ comparing the values at the end of the study with the baseline.

## 4. Discussion

The present study investigated impulsive personality traits in individuals with T2D as predictors of BMI changes and diabetes control, measured by the levels of glycated hemoglobin after 3 years of lifestyle interventions. 

Baseline evaluations of metabolic parameters and personality traits showed differences between individuals with T2D and those without it. As expected, higher levels of glucose, HOMA-IR and HbA_1c_ were characteristic of individuals with T2D. Concerning impulsive personality traits: although the UPPS scores were sub-clinical in both groups, patients with T2D showed significantly higher total scores and positive urgency scores at baseline. Positive urgency is defined as the tendency to act rashly in response to positive affective states [35], which has been strongly associated with emotional factors [32]. Neuroimaging studies have suggested that emotional-based dispositions to act impulsively are linked to orbitofrontal cortex–amygdala connections in the brain, regulated by dopamine and serotonin neurotransmitters [30]. Individuals with T2D showed impaired central insulin signaling, which has a direct impact on brain dopamine systems [15,36,37,38], indirectly contributing to impulsivity. This is exemplified in the impulsive decision making shown in individuals with T2D [13,14]. According to this view, the current results suggest that impulsive tendencies in individuals with T2D may be particularly affected during emotional states, mainly those involving positive emotions.

When focusing on the T2D group, no correlations at baseline were shown between the impulsivity measures and metabolic parameters. However, a higher UPPS-P total score at baseline predicted higher weight at follow-up. Likewise, baseline scores in sensation seeking predicted a lower decrease in weight/BMI after 3 years of lifestyle intervention. This result could be explained by the link between sensation seeking and behaviors associated with the prospect of reward or novel stimuli [39], including high rewarding foods. Furthermore, a higher BMI at 3 years was predicted by baseline scores in lack of perseverance. This reflects a reduced ability to remain focused on difficult and boring tasks. Associations between lack of perseverance and BMI have been described in transversal studies in individuals with obesity [40]. The present results suggest that low perseverance affects weight control over time in individuals with T2D. These findings could have significant implications for T2D management and prevention interventions, since overweight and obesity represent major risk factors for a poor T2D prognosis and the development of diabetic complications [2]. Lifestyle interventions in older adults have shown significant reductions in body weight and HbA_1c_ [41], so knowing a differential prognosis in the presence of impulsivity traits could help in the planning of a more effective therapeutic intervention or treatment. 

Concerning diabetes management, in our population with T2D, impulsive traits were not predictors of glycemic control after 3-years of follow-up. By contrast, another study found that impulsivity negatively correlated with diabetes self-management and positively correlated with self-reported HbA_1c_ levels during a three-months period [22]. However, discrepancies in the results could be related by different timings for the assessment of the HbA_1c,_ with a three-year follow-up considered in the present study. This would suggest that even though impulsive personality traits may correlate with glycemic control, it may not have a long-term effect on the control of diabetes. These results, however, must be interpreted with caution, as our sample presented sub-clinical levels of impulsivity. Therefore, it might be expected that, in individuals with T2D characterized by high impulsivity, this personality trait could have a negative effect on diabetes control. The results of this study should be interpreted in light of some limitations. Firstly, even though the statistical model was adjusted for the intervention group and other potential confounders to diminish the residual effect, we cannot completely discard that these factors are affecting the observed associations. However, it is important to acknowledge that the individuals randomized into the “control” group are also recommended to follow a healthy Mediterranean diet that has previously shown beneficial effects [42,43]. A second limitation is related to the generalizability of these results. For instance, given the samples’ sub-clinical scores in the UPPS-P and the moderate statistical effect, results should be interpreted with caution. To overcome this limitation, future studies in individuals with T2D presenting highly impulsive traits should be conducted. Thirdly, it should be mentioned that self-report measures are not free from response bias, and a more comprehensive evaluation of impulsivity using more objective measures (e.g., cognitive tests) could be useful to better characterize impulsive profiles in individuals with T2D. Additionally, given that our population represent a sample of older Mediterranean adults with metabolic syndrome, further research is needed to determine the presence of these associations in a younger sample without the coexistence of other metabolic disorders. Despite these limitations, several strengths should be highlighted, including the length of the follow-up (i.e., 3 years) that was conducted in a large sample of individuals with T2D. In clinical terms, these results suggest the importance of considering impulsivity as an additional factor that may negatively affect patients’ adherence, opening future research to the development of ad hoc intervention for individuals with T2D characterized by more impulsive personality traits. 

## 5. Conclusions

Overall, our study results showed that, in this older adult population, there are differences in personality traits in individuals with or without T2D, where UPPS-P scores, positive urgency and total factor registered a higher mean within individuals with T2D, indicating that higher scores on the impulsivity scale are associated with T2D, and these are predictors of weight loss but not of diabetes control after 3 years of lifestyle intervention. Further research is required to determine whether a clinic impulsive trait could have an effect on diabetes control. 

## Figures and Tables

**Table 1 jcm-11-03476-t001:** Descriptive of the sample at baseline.

	Total (*n* = 447)		T2D (−)(*n* = 308)		T2D(+) (*n* = 139)		
	*n*	%	*n*	%	*n*	%	*p*
Sex							
*Men*	216	48.3%	146	47.4%	70	50.4%	0.563
*Women*	231	51.7%	162	52.6%	69	49.6%	
Civil status							
*Single*	17	3.8%	12	3.9%	5	3.6%	0.886
*Married*	351	78.5%	244	79.2%	107	77.0%	
*Divorced-separated*	29	6.5%	20	6.5%	9	6.5%	
*Widowed*	50	11.2%	32	10.4%	18	12.9%	
School							
*University (high)*	42	9.4%	30	9.7%	12	8.6%	0.985
*University (grade)*	36	8.1%	25	8.1%	11	7.9%	
*Secondary*	130	29.1%	89	28.9%	41	29.5%	
*Primary*	239	53.5%	164	53.2%	75	54.0%	
Employment							
*Unemployed*	81	18.1%	61	19.8%	20	14.4%	0.307
*Work at home*	47	10.5%	35	11.4%	12	8.6%	
*Retired*	287	64.2%	189	61.4%	98	70.5%	
*Unemployed*	32	7.2%	23	7.5%	9	6.5%	
Group weight							
*Overweight*	124	27.7%	91	29.5%	33	23.7%	0.145
*Obesity I (BMI 30–35)*	217	48.5%	150	48.7%	67	48.2%	
*Obesity II (BMI 35–40)*	102	22.8%	63	20.5%	39	28.1%	
*Obesity III (BMI > 40)*	4	0.9%	4	1.3%	0	0.0%	
Intervention group							
*Control*	228	51.0%	161	70.6%	67	29.4%	0.425
*Experimental*	219	49.0%	147	67.1%	72	32.9%	
	*Mean*	*SD*	*Mean*	*SD*	*Mean*	*SD*	*p*
Age (years-old)	65.19	4.60	64.83	4.59	65.97	4.53	**0.015 ***
	*Mean*	*SD*	*Mean*	*SD*	*Mean*	*SD*	*p*
HOMA-IR	5.43	3.05	4.86	2.51	6.69	3.71	**<0.001 ***
Glucose	116.60	29.68	103.61	12.95	145.37	35.55	**<0.001 ***
Glycated hemoglobin (HbA_1c_)	6.15	0.78	5.80	0.39	6.91	0.87	**<0.001 ***
	Mean	SD	Mean	SD	Mean	SD	p
UPPS-P Lack of premeditation	19.85	5.60	19.69	5.59	20.18	5.62	0.397
UPPS-P Lack of perseverance	18.92	5.01	18.83	5.06	19.14	4.89	0.547
UPPS-P Sensation seeking	20.51	7.14	20.29	7.22	20.99	6.97	0.338
UPPS-P Positive urgency	23.10	8.28	22.52	8.08	24.38	8.60	**0.028 ***
UPPS-P Negative urgency	24.57	7.35	24.21	7.11	25.38	7.84	0.118
UPPS-P Total score	106.95	22.62	105.55	22.40	110.07	22.86	**0.049 ***

Note. T2D(−): diabetes absent. T2D(+): diabetes present. SD: standard deviation. BMI: body mass index. HOMA-IR: Homeostasis Model Assessment of Insulin Resistance index. UPPS-P: impulsive behavior scale. * Bold: significant comparison.

**Table 2 jcm-11-03476-t002:** Significant predictors at 3 year follow up, considering impulsivity measures at baseline; final models adjusted by sex, age, education and intervention group (results for the subsample of patients with T2D).

Criteria (3 Years)	UPPS-P (at Baseline)	B	SE	Beta	*p*	95%CI B	
BMI	Lack of perseverance	0.122	0.064	0.160	0.048	0.005	0.250
Changes in BMI	Sensation seeking	−0.067	0.022	−0.259	0.002	−0.110	−0.025
Weight	Total score	0.125	0.048	0.196	0.010	0.030	0.219
Changes in weight	Sensation seeking	−0.218	0.058	−0.305	<0.001	−0.333	−0.103
Hb1Ac	*No significant predictors*						
Changes in Hb1Ac	*No significant predictors*						

Note. UPPS-P: impulsive behavior scale. B: non-standardized coefficient. SE: error standard. Beta: standardized coefficient. 95%CI: 95% confidence interval. BMI: body mass index. HbA1c: Glycated hemoglobin.

## Data Availability

Due to signed consent agreements regarding data sharing, there are restrictions on data availability for the PREDIMED-Plus trial. These only allow access to external researchers for studies following the project purposes. Requestors wishing to access the PREDIMED-Plus trial data used in this study can make a request to the PREDIMED-Plus trial Steering Committee chair: jordi.salas@urv.cat. The request will then be passed to members of the PREDIMED-Plus Steering Committee for deliberation.

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
