# Peer review of "Impulsive Personality Traits Predicted Weight Loss in Individuals with Type 2 Diabetes after 3 Years of Lifestyle Interventions"

_jcm, 2022, doi:10.3390/jcm11123476_

Round 1

Reviewer 1 Report

This is a longitudinal study to determine the impulsive personality trait as a predictor of weight loss in people with T2D after 3 years of lifestyle intervention.

It is identified that the title and the abstract talk about people with T2D and the results indicate comparisons of people with T2D and without T2D. It should describe in the introduction why they should obtain different results when presenting or not diagnosing with T2D.

The results of this work are part of a 3-year Mediterranean diet intervention. How is it possible to explain that the weight loss results are due to impulsivity and not to the Mediterranean diet?

A sample of 139 patients with T2D is reported. What percentage of the participants were in the control and intervention group? No information on this is mentioned. Because the statistical analyzes were adjusted for sex, age, education and intervention group. So the final sample size should be smaller. Hence, the results shown have implications due to the size of the sample when considering a longitudinal study.

The following results do not respond to any objective:

Within the T2(+) condition, changes between T0 and T3 (adjusted by sex, age, educa-189 tion and intervention group) were significant for Hb1Ac (T0: mean = 6.92, SD = 0.87; T3: 190 mean = 6.74, SD = 1.00; p = 0.027, n2 = 0.037), weight (T0: mean = 87.71, SD = 14.50; T3: mean 191 = 83.72, SD = 14.16; p < 0.001, n2 = 0.427) and BMI (T0: mean = 32.84, SD = 3.40; T3: mean = 192 31.48, SD = 3.61; p < 0.001, n2 = 0.397).

Author Response

This is a longitudinal study to determine the impulsive personality trait as a predictor of weight loss in people with T2D after 3 years of lifestyle intervention.

It is identified that the title and the abstract talk about people with T2D and the results indicate comparisons of people with T2D and without T2D. It should describe in the introduction why they should obtain different results when presenting or not diagnosing with T2D.

Response: Thank you for this recommendation. Differences in impulsivity in terms of cognitive functions have been shown in T2D individuals by previous works reported in the introduction. On the same line, it is expected that more impulsive personality traits could be also found in relation to the presence of T2D. We have reported this link in the introduction section (lines: 77-79). 

The results of this work are part of a 3-year Mediterranean diet intervention. How is it possible to explain that the weight loss results are due to impulsivity and not to the Mediterranean diet?

Response: Thank you for this constructive remark. Sorry for the misunderstanding, we agree with the reviewer that the intervention and the MedDiet indications given to the control group are responsible for the weight loss, not the impulsivity personality traits. Our hypothesis was formulated assuming this, and that higher levels of impulsivity may be negative affecting the results of the intervention. We tried to better explain our main idea throughout the manuscript.

A sample of 139 patients with T2D is reported. What percentage of the participants were in the control and intervention group? No information on this is mentioned. Because the statistical analyzes were adjusted for sex, age, education and intervention group. So the final sample size should be smaller. Hence, the results shown have implications due to the size of the sample when considering a longitudinal study.

Response: We appreciate this thoughtful feedback provided from the Reviewer. We have now included in table 1 the percentages of the participants in each group.
We apologize for the misunderstanding, the analyses were statistically adjusted for the intervention group for the total sample of T2D (n=139), so it didn't have any implications in the sample size of the longitudinal study.  

The following results do not respond to any objective:

Within the T2(+) condition, changes between T0 and T3 (adjusted by sex, age, educa-189 tion and intervention group) were significant for Hb1Ac (T0: mean = 6.92, SD = 0.87; T3: 190 mean = 6.74, SD = 1.00; p = 0.027, n2 = 0.037), weight (T0: mean = 87.71, SD = 14.50; T3: mean 191 = 83.72, SD = 14.16; p < 0.001, n2 = 0.427) and BMI (T0: mean = 32.84, SD = 3.40; T3: mean = 192 31.48, SD = 3.61; p < 0.001, n2 = 0.397).

Response: Thank you for this constructive remark. We agree with the reviewer and have now eliminated this paragraph of the manuscript. 

Reviewer 2 Report

The manuscript reports an interesting and well-described multicenter study, with a large number of participants about the relationships between T2D and impulsivity. The study has several merits like the implementation of the UPPS and the number of participants. The manuscript is clear and the methods are corrected. My suggestions for the authors are:

- do you think that the assessment of impulsivity with a self-report questionnaire could bias the results? Is it possible that different methodologies could find other significant results?

- is it possible that also other elements could have a role in impulsivity rates (for example vitamin D or inflammatory cytokines)?

- overall discussion seems to be very limited to the explanation of the missed results. I think the paper would allow the authors to draft possible future trajectories of research with supporting data. Please consider an implementation of the section.

- have you evaluated the difference between the follow-up timing in the literature and in your study? Maybe it could explain partially your missing results. 

Author Response

The manuscript reports an interesting and well-described multicenter study, with a large number of participants about the relationships between T2D and impulsivity. The study has several merits like the implementation of the UPPS and the number of participants. The manuscript is clear and the methods are corrected. My suggestions for the authors are:

- do you think that the assessment of impulsivity with a self-report questionnaire could bias the results? Is it possible that different methodologies could find other significant results?

Response: We thank the reviewer for highlighting this point. As you suggested, self-report measures could produce a sort of bias in the results compared to more objective measures, such as the cognitive ones. This concept has been added to the limitations section (lines: 268-271). However, impulsivity is a multidimensional construct and here we were interested in investigating impulsivity traits rather than other domains of impulsivity (e.g., decision-making, inhibitory control, etc.). In order to do so, we adopted the UPPS, which is a recognised measure of personality traits related to impulsivity. 

- is it possible that also other elements could have a role in impulsivity rates (for example vitamin D or inflammatory cytokines)?

Response: We find the reviewer's question very interesting. We agree that aspects linked to vitamin D or inflammatory status, as other factors that can be affecting the insulin signalling among others, could have a role in impulsivity rates. In this analysis, however, we did not consider the variables that are affecting impulsivity, but rather how impulsivity rates are affecting the diabetes control and the weight-loss process. It will be a very interesting approach for our future lines, maybe considering cognitive impulsivity, as has more change variability than impulsivity personality traits, that trend to remain unchanged. 

- overall discussion seems to be very limited to the explanation of the missed results. I think the paper would allow the authors to draft possible future trajectories of research with supporting data. Please consider an implementation of the section.

Response: Thank you for this recommendation. As you suggested, we mentioned the possible clinical implication of this study and future directions that could derive from the present results (lines: 274-280). 

- have you evaluated the difference between the follow-up timing in the literature and in your study? Maybe it could explain partially your missing results. 

We thank the reviewer for highlighting this point. As suggested, the follow-up timing considered in previous literature is shorter than the one from our study. We agreed that this is an important point potentially explaining differences in the outcomes. We have now added this important information to the discussion (lines: 251-253). 

Round 2

Reviewer 1 Report

I suggest, the data of people who do not have diabetes should be eliminated in the methodology and results.

Author Response

I suggest, the data of people who do not have diabetes should be eliminated in the methodology and results.

RESPONSE: Thank you very much for your suggestion. We agree with you that it can generate confusion to include the entire sample in our first results. Our intention is to be able to show the reader that the reason we specifically analysed predictors in the diabetic sample is because they showed higher trait impulsivity scores than those individuals without diabetes. These results act as a form of justification for our subsequent analyses. In addition, this is the first study to compare diabetic and non-diabetic individuals according to the UPPS model, which conceptualizes the multivariate structure of impulsivity leading to a better characterisation of the trait dimensions of impulsivity in this population. Given these reasons, we considerate appropriate to keep the data from the analyses in both diabetic and non-diabetic participants.

This manuscript is a resubmission of an earlier submission. The following is a list of the peer review reports and author responses from that submission.

Round 1

Reviewer 1 Report

This paper explores the link between impulsivity and weight management in people with type 2 diabetes.  I am a psychologist, unfamiliar with the clinical literature – though I can comment on the assessment of impulsivity, the design, analysis and interpretation of the results.

The Introduction could be clearer, and language/grammatical issues need to be addressed – for example:-

  • “This increment was especially observed in older adults, where individuals with T2D aged 65 or older constitute about half of the total population with this condition”. Why not say “about half of those with T2D are aged 65 or older”?
  • The spelling of Barratt on Line 52
  • “increase” should be “increased” on line 61
  • “with overweight/obese” should read “who were overweight/obese” line 69
  • Clumsy phrasing lines 70-76

  The authors make the following points in the following order in Lines 58-68, but I had major problems understanding how making these points led to a coherent research hypothesis.

  • Impulsivity is related to weight/weight gain. A great many studies show this, and the underlying mechanisms are starting to be understood.  See Aiello et al (2018) for a summary.
  • Little is known about the relationship between impulsivity and T2D
  • T2D has increased worldwide
  • Half the cases are aged 65+
  • Aging affects T2D (including changes in muscle & fat)
  • “T2D increases the risk of comorbidities such as cognitive decline… Several studies have explored the association between executive
    functions and BMI in elderly” (66-68).  However the present authors do not mention that there are conflicting results in the elderly, with some reporting reduced executive functions whilst others show opposite effects (Dahl et al., 2010; Smith, Hay, Campbell, & Trollor, 2011).
  • However little is known about impulsivity in these cases.
  • A X-sectional study shows that overweight people with T2D show a form of impulsivity is related to BMI (70). Longitudinal studies in obese people shows that impulsivity predicts weight gain.  (73)
  • Other psychological variables (of little relevance here) predict weight loss following bariatric surgery
  • “Still, the relationship between T2D and impulsive traits and its effects on weight loss have not been investigated”

I really think this introduction needs to be completely re-written, more tightly focused, and trimmed of irrelevant material. 

There is no point in breaking down the sample by origin – there are only 8 South American participants.  Drop them?

I cannot see the relevance of some of the analyses, given the stated aim of the paper.

For example, Table 2 and Figure 1 show how impulsivity changed for the diabetic and non-diabetic groups from the baseline to 1-year and 3-year assessments.  Why is this thought to be important? The purpose of the study is not to explore whether impulsivity changes over time, but to determine how impulsivity influences weight/weight-loss.  The effect sizes reported here are TINY (though statistically significant) reflecting differences of about a seventh of a standard deviation.  and the authors do not mention/discuss this. The Y-axis of Figure 1 needs a legend, it should show error-bars, and I cannot understand where the extra data point half way between T1 and T3 comes from.  Something is wrong here.

The use of stepwise regressions is never appropriate (lines 194-207).  See for example Harrell (2015) and Miller (2002).  As this analysis is the most important part of the paper, this is a major problem.

The Discussion needs to discuss effect sizes; are the relationships large enough to be of interest or clinical value?

References

Aiello, M., Ambron, E., Situlin, R., Foroni, F., Biolo, G., & Rumiati, R. I. (2018). Body weight and its association with impulsivity in middle and old age individuals. Brain and Cognition, 123, 103-109. doi:10.1016/j.bandc.2018.03.006

Dahl, A., Hassing, L. B., Fransson, E., Berg, S., Gatz, M., Reynolds, C. A., & Pedersen, N. L. (2010). Being overweight in midlife is associated with lower cognitive ability and steeper cognitive decline in late life. J Gerontol A Biol Sci Med Sci, 65(1), 57-62. doi:10.1093/gerona/glp035

Harrell, F. E., Jr. (2015). Regression Modeling Strategies (2 ed.). New York: Springer.

Miller, A. J. (2002). Subset selection in regression. London: Chapman & Hall.

Smith, E., Hay, P., Campbell, L., & Trollor, J. N. (2011). A review of the association between obesity and cognitive function across the lifespan: implications for novel approaches to prevention and treatment. Obesity Reviews, 12(9), 740-755. doi:10.1111/j.1467-789X.2011.00920.x

Reviewer 2 Report

I appreciate the opportunity to review the work entitled “Impulsive personality traits predicted weight loss in individuals with type 2 diabetes after 3 years of lifestyle interventions”.

Throughout the document there are inconsistencies and lack of justification for the work. The abstract mentions that the study focuses on weight control and glycemia management in people with T2D, but the methodology refers to people with MetS.

In the introduction, in the first paragraph, two questionnaires are described to assess the trait of impulsiveness without any relevance (Lines 52 to 57). In the second paragraph, there is no consistency in the ideas. The aging process is described as a vulnerability to developing diabetes and the paragraph ends with the lack of studies that address the impulsivity trait (lines 63 to 68). In paragraph 3, the objective is different from that shown in the abstract. The objective focuses on comparing impulsive personality traits in individuals meeting metabolic syndrome (MetS) criteria with or without T2D. In the abstract: The study aimed at investigating traits impulsivity in T2D population and its predicting role in long-term weight control and glycemic management. The introduction should be focused on the importance of weight loss and glycemic control in people with T2D or MetS.

Methods

On lines 99 to 101, the inclusion criteria, it is mentioned to have MetS (3 to 5 risk factors), but, the title of the work mentions only people with DT2. The number of risk factors (MetS) may have implications for weight loss and glycemic control. In lines 103 to 108, it is mentioned that the total sample was randomized and distributed in the intervention and control group. But no information is reported about the sample being used in this study. Are they from the control or intervention group? This information has implications for the results.

Results

On lines 174-175. It is described about a null hypothesis, but it was not mentioned before about hypotheses.

Lines 178 to 185. The changes in the impulsivity score in the different evaluations (T0, T1, and T3) are described. Did the intervention influence these changes? What is the purpose of displaying these results? Where are the results on weight loss and HbA1c (T0, T1, and T3)?

Lines 194 to 201. The results shown in table 3 are confusing. Are the results shown in Table 3 from the baseline analysis or follow-up evaluation? Is the sample size the same in the Baseline evaluation and the 3-year follow-up?

In general, the document presents various inconsistencies: lack of justification in the introduction and population of interest, imprecision in the methodology, and lack of clarity in the results. The authors omit important information on the variables of interest (weight loss and HbA1c). The results shown in this document are part of an intervention study. How is it justified that weight loss is related to the impulsivity trait and not to the intervention?